# Ludax: A GPU-Accelerated Domain Specific Language for Board Games

## Abstract

Games have long been used as benchmarks and testing environments for research in artificial intelligence. A key step in supporting this research was the development of *game description languages*: frameworks that compile domain-specific code into playable and simulatable game environments, allowing researchers to generalize their algorithms and approaches across multiple games without having to manually implement each one. More recently, progress in reinforcement learning (RL) has been largely driven by advances in *hardware acceleration*. Libraries like JAX allow practitioners to take full advantage of cutting-edge computing hardware, often speeding up training and testing by orders of magnitude. Here, we present a synthesis of these strands of research: a domain-specific language for board games which automatically compiles into hardware-accelerated code. Our framework, `Ludax`, combines the generality of game description languages with the speed of modern parallel processing hardware and is designed to fit neatly into existing deep learning pipelines. We envision `Ludax` as a tool to help accelerate games research generally, from RL to cognitive science, by enabling rapid simulation and providing a flexible representation scheme. We present a detailed breakdown of `Ludax`'s description language and technical notes on the compilation process, along with speed benchmarking and a demonstration of training RL agents.

## 1 Introduction

For the past 75 years, games have served as vital tests and benchmarks for artificial intelligence research. While many specific games have been completely solved (Schaeffer et al., 2007) or optimized beyond the abilities of the strongest human players (Campbell et al., 2002; Silver et al., 2017), the general space of games remains a fertile ground for measuring improvements in reasoning, planning, and strategic thinking. A critical part of this progress, however, is the ability to test approaches and algorithms on a set of environments that are both diverse and computationally efficient.

To help drive further games and learning research, we introduce `Ludax`: a domain-specific language for board games that compiles into GPU-accelerated code written in the JAX library (Bradbury et al., 2018). `Ludax` draws on two main inspirations: (1) `Ludii` (Piette et al., 2020), a general purpose description language for board games capable of representing more than 1400 games from throughout history and around the world, and (2) `PGX` (Koyamada et al., 2023), a collection of optimized JAX-native implementations of classic board games and video games designed to facilitate rapid training and evaluation of modern reinforcement learning (RL) agents. `Ludax` presents a flexible and general-purpose game representation format that can be leveraged for efficient simulation and learning on modern computing hardware.

`Ludax` currently supports two-player, perfect-information, turn-based board games played by placing, capturing, and moving pieces. This set of mechanics is broad enough to capture a wide range of existing games (e.g. *Connect Four*, *Pente*, *Hex*, ...) as well as many unexplored *novel* games and variants that fall within that class. Further, `Ludax` is designed to be easily expandable – like with `Ludii`, implementing new game mechanics in `Ludax` only requires implementing new atomic components in the underlying description language. These components can then be combined compositionally with existing elements of the language to produce an entirely new *range* of possible games, instead of each game needing to be implemented separately.

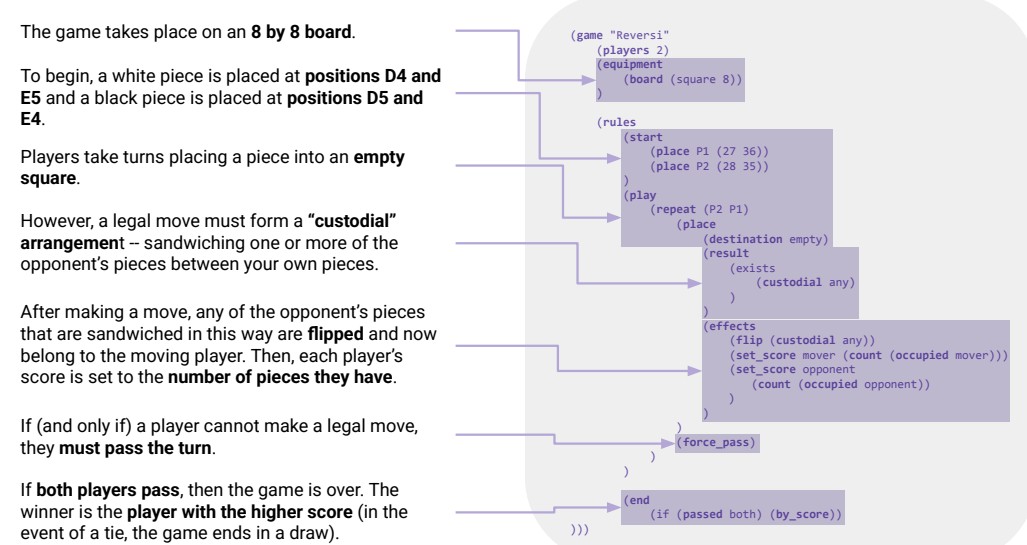

The game takes place on an **8 by 8 board**.

To begin, a white piece is placed at **positions D4 and E5** and a black piece is placed at **positions D5 and E4**.

Players take turns placing a piece into an **empty square**.

However, a legal move must form a **"custodial" arrangement** -- sandwiching one or more of the opponent's pieces between your own pieces.

After making a move, any of the opponent's pieces that are sandwiched in this way are **flipped** and now belong to the moving player. Then, each player's score is set to the **number of pieces they have**.

If (and only if) a player cannot make a legal move, they **must pass the turn**.

If **both players pass**, then the game is over. The winner is the **player with the higher score** (in the event of a tie, the game ends in a draw).

Figure 1: **Natural language description of *Reversi* along with its corresponding translation into Ludax**. Ludax uses "ludemic" syntax that represents high-level game components as separate program sections and aims to be easily interpretable to non-experts.

Another design goal for `Ludax` is ease of use, both in terms of game design and experimentation. The syntax of the description language is "ludemic" (Piette et al., 2020) – splitting game rules into clear sections governing the game's setup, play mechanics, and end conditions. Like with `Ludii`, game programs in `Ludax` resemble English descriptions of rules (see Figure 1). Further, by leveraging the structure of the existing `PGX` library, environments instantiated in `Ludax` can be easily combined with existing frameworks for GPU-accelerated search, reinforcement learning, or evolution (DeepMind et al., 2020; Tang et al., 2022). `Ludax` also supports a basic web interface for interactive debugging and potential user-studies.

`Ludax` is fundamentally a platform for accelerating board game research. In an era of increasingly complicated tasks and benchmarks, relatively simple board games may seem to be less interesting research domains (especially as many games have been more-or-less "solved" by modern methods). However, `Ludax` is not just a collection of new tasks. By decoupling rapid execution from the intensive process of writing new environment code, `Ludax` can power new research in a variety of directions. For instance, `Ludax` can be used to analyze RL generalization (Soemers et al., 2025) by defining a wide range of modifications for a target game task (akin to a platform like `Minihack` (Samvelyan et al., 2021)) or help improve studies of game generation by enabling the rapid evaluation of procedurally-generated rulesets (Todd et al., 2024; Collins et al., 2025). Finally, `Ludax` can help advance recent research into world modeling (Ying et al., 2025) by (1) providing a wide and easily-refreshable set of environments to test on efficiently and (2) allowing automated systems to propose and refine world models in these "novel games" by writing high-level and semantically-meaningful DSL code.

To our knowledge, `Ludax` is the first board game description language which compiles into GPU-accelerated code. In the following sections, we provide a detailed description of the language syntax, compilation process, and `Ludax`'s expressive range. We also provide speed benchmarking compared to both `Ludii` and `PGX`, as well as an initial demonstration of training learned agents. Finally, we conclude with a discussion of potential use cases and future directions.

## 2    RELATED WORK

**Game Description Languages:**  Game description languages have been used for many years and in a variety of domains. The Stanford GDL (Love et al., 2008; Genesereth & Thielscher, 2014;

Schiffel & Thielscher, 2014; Thielscher, 2017) is among the most influential, helping to popularize research in general game playing (Pitrat, 1968) through its use in the International General Game Playing Competition (Genesereth et al., 2005; Genesereth & Björnsson, 2013). Other notable examples include `VGDL` (Ebner et al., 2013; Schaul, 2013; 2014) (primarily known from its use in the General Video Game AI framework (Perez-Liebana et al., 2019)), `RBG` (Kowalski et al., 2019), `Ludi` (Browne, 2009), and its successor `Ludii` (Piette et al., 2020). GDLs have also been used to describe the rules of card games (Font et al., 2013) as well as to represent human goals in naturalistic simulated environments (Davidson et al., 2022; 2025). Modern game description languages have tended to move away from a basis in formal logic in favor of greater human usability, though there are benefits in efficiency gained by the use of regular languages (Kowalksi et al., 2020).

**GPU-Accelerated Environments:**  Recent years have seen a proliferation of learning environments implemented in the JAX library or other frameworks that enable hardware (typically GPU) acceleration. Examples include single-agent and multi-agent physics simulators (Freeman et al., 2021; Makoviychuk et al., 2021; Bettini et al., 2022), ports of both classic and recent reinforcement learning tasks (Dalton & Frosio, 2020; Lange, 2022; Koyamada et al., 2023; Matthews et al., 2024), combinatorial optimization problems (Bonnet et al., 2023), multi-agent coordination problems (Rutherford et al., 2023), and driving simulators (Gulino et al., 2023; Kazemkhani et al., 2025). While these efforts have spurred significant progress and span a wide range of domains and task formulations, each of them implement a fixed environment or set of environments. As such, they cannot easily be extended to novel environments without first writing new hardware-accelerated code. `Ludax` stands alongside a number of description languages for other domains (e.g. probabilistic programming, planning, single-player puzzles) that leverage JAX for efficient execution (Chandra et al., 2025; Gimelfarb et al., 2024; Earle & Togelius, 2025; Earle et al., 2025).

## 3 DESCRIPTION LANGUAGE DETAILS

`Ludax`'s game description language draws heavily on the `Ludii` description language, particularly in its use of "ludemic" syntax that represents game rules in terms of high-level and easily-understandable components (Piette et al., 2020). The complete grammar file and syntax details are available in the Supplemental Material.

### 3.1 EQUIPMENT AND START RULES

The `equipment` section contains information about the physical components used by the game. Currently, this only specifies the size and shape of the board (i.e. whether it is square, rectangular, hexagonal, or hexagonal-rectangular). The dimensions and shape of the board are used during compilation to help pre-compute certain game-relevant properties, such as the board indices corresponding to lines of specific lengths. In future versions of `Ludax`, the equipment section will also detail the different pieces used by each player if the game specifies more than one.

The `start` section is an optional section that contains the rules for the game's setup. For most games, play begins on an empty board and the `start` section is omitted. In some games, such as *Reversi* (see Figure 1), pieces are placed in a particular arrangement at the start of play.

### 3.2 PLAY RULES

Typically, the `play` rules of each game are the most involved, as they detail the core mechanics and dynamics of the game. The `play` section is itself broken into one or more subsections called "play phases." Each phase has its own rules for player actions and turn-taking, as well as specific conditions for when to transition to another phase. Most games have only a single phase in which players alternate turns until the game is over, specified with the `repeat` keyword. Some games include a `once_through` phase that progresses through the turn order a single time before advancing to the next phase. The sequence of player turns is specified independently for each phase. For instance, *Yavalax* (Appendix Figure 4, bottom-left panel) begins with the first player making a single move (i.e. `(once_through (P1) ...)` ) before both players alternate taking two turns for the rest of the game (i.e. `(repeat (P2 P2 P1 P1) ...)` ).

The core of each phase is a "play mechanic" that encodes the ways that players take their turns. In the context of reinforcement learning, a play mechanic specifies both the action space ($\mathcal{A}$) and the transition function ($\mathcal{T} : \mathcal{S} \times \mathcal{A} \rightarrow \mathcal{S}$). At a lower level, each play mechanic also defines a "legal action mask function" that returns whether each action is valid from the current game state. Currently, `Ludax` supports only one kind of play mechanic: `place`. A `place` mechanic's primary argument is a `destination` constraint which specifies where a piece may be placed on a given player's turn. For many games, such as *Tic-Tac-Toe*, this is simply the set of empty board positions. For some games, however, the destination constraint is more involved: in *Connect Four* (Appendix Figure 4, top-right panel), legal actions are empty spaces that are on the bottom edge of the board or immediately above an occupied position (see subsection 3.4 for a discussion of how actions are represented more generally in `Ludax`). Even further, some games have what we call `result` constraints which require that a legal action results or doesn't result in a board with specific properties. *Yavalax* and *Reversi* both use `result` constraints: the former forbids players from placing a piece that forms a line of five or that forms only a single line of four, whereas the latter requires players to place a piece in a way that "sandwiches" one or more of their opponent's pieces in a line. Finally, a play mechanic may optionally specify one or more `effects` that modify the game state after the action is performed. Effects are used to handle mechanics like capturing or flipping pieces, as well as updating each player's score (if the game uses score). Both *Reversi* and *Pente* (Appendix Figure 4, bottom-right panel) use play effects to handle flipping and capturing pieces, respectively, with *Pente* also using the score as an alternate winning condition.

Throughout this section, we have been referring to various properties of a game state and relationships between pieces / positions (e.g. whether pieces are "sandwiched," whether a line is formed, whether a piece is adjacent to another, ...). These are the lowest-level component's of `Ludax`'s description language and are referred to collectively as `masks`, `functions`, and `predicates`. A `mask` takes in the current game state and returns a boolean array over each position on the board. Some masks, like `occupied` or `edge`, take additional grammatical arguments which might specify a particular player or region of the board.[1] A `function` similarly takes in the current game state and returns a single non-negative integer. In `Ludax`'s current form, `line` is probably the most commonly-used `function` – it returns the number of contiguous lines of a given player's pieces on the board, with a specified length and orientation. Lastly, a `predicate` maps from a game state to a single boolean truth value. Many predicates operate over the outputs of `masks` and `functions`, such as `exists` or `equals`, though some like `mover_is` are computed directly from game states. Crucially, the outputs of `masks`, `functions`, and `predicates` can be combined compositionally using first-order logic (excluding quantification) to form more complicated expressions. So, the condition "*if Player 2 makes a line of 4 in a row or a diagonal line of 3...*" would be rendered as follows:

```
(and (mover_is P2) (or (line 4) (line 3 orientation:diagonal)))
```

Note that, for ease of use, `Ludax` automatically interprets the presence of a bare `function` inside a boolean operator as indicating a non-zero value. So, `(line 4)` is equivalent to `(>= (line 4) 1)`.

## 3.3 END RULES

The last section of a game description in `Ludax` details the criteria that terminate a game. The `end` section contains one or more "end conditions" – these are applied *in order*, with the first condition to activate determining the ending behavior (i.e. which player wins or if the game ends in a draw). If none of the conditions activate, then the game continues. For instance, *Tic-Tac-Toe* includes both the end conditions `(if (line 3) (mover win))` and `(if (full_board) (draw))`, with the draw condition only triggering if the "three in a row condition" is not met. End conditions also frequently refer to a player's score, which is updated or set as a result of an action's effects (see above).

---

[1] The `adjacent` mask is a special case – it takes *another* mask as an additional argument and returns board positions adjacent to any of the active positions in the original mask.

## 3.4 Design Considerations

While `Ludax` draws heavily from the `Ludii` description language, there are some important differences which go beyond just changes in syntax. The first of these relates to how both systems represent a game's action space. One of the design goals of `Ludii` is that game descriptions should resemble as much as possible the rules in natural language. In *Connect Four*, for instance, players take a move by dropping a piece into one of seven columns of the board, at which point the piece falls until it reaches the bottom or rests on another piece. Accordingly, the canonical representation of *Connect Four* in `Ludii` features pieces that "`Drop`" into the "`LastColumn`" chosen by the player (`PGX` implicitly represents the game in a similar way). As mentioned above, however, `Ludax` represents the action space differently: players simply place a piece onto an empty board cell, with actions that are not directly above an existing piece or the bottom of the board marked as illegal. Mechanically, the two implementations of *Connect Four* are identical – the difference lies in how they are encoded (especially to simulated players or reinforcement learning agents). The "column-based" representation has many advantages (it matches the physical properties of the game in real life and lowers the branching factor), but it is also *game-specific*. While `Ludax` also strives to represent game descriptions intuitively, we primarily aim to provide a unified representation format across games, such that general game-playing agents can more easily transfer knowledge and expertise from one game to another. As such, the size and form of the action space for any `place`-based game is determined only by the size and shape of the board. This choice is also partially motivated by the specifics of working with the JAX library (see Section 4) and has implications for benchmarking and downstream use-cases (see Section 6).

## 4 Compiling Game Descriptions into Game Environments

In this section, we describe the high-level approach used to map from programs in the `Ludax` game description language to hardware-accelerated simulation environments. While `Ludax` specifically instantiates board game environments using the `Lark` Python library, the general approach is flexible enough to be used with different domains and parsing toolkits. Broadly speaking, `Ludax` operates by defining the leaves of the grammatical parse tree (i.e. individual `masks`, `functions`, and `predicates`) as atomic functions written in JAX, which are then dynamically composed from the bottom-up to form higher-level operators used by the environment class. Consider again the following game expression:

```
(and (mover_is P2) (or (line 4) (line 3 orientation:diagonal)))
```

During compilation, the leaf-level nodes (i.e. `(mover_is P2)` and `(line 4)`) are converted into JAX functions which map from the current game state to (in this case) a boolean truth value, and those functions are then passed up the parse tree. Higher-level nodes, such as `(and ...)`, receive the JAX functions corresponding to each of their children and return a *new* JAX function that also takes the game state as input and implements the appropriate operation (in this case, boolean conjunction). In pseudocode, using the `Lark` library's `Transformer` paradigm, this looks like the following:

```
def predicate_and(self, children):
    def predicate_fn(state):
        children_values = [child_fn(state) for child_fn in children]
        return all(children_values)

    return predicate_fn
```

In actuality, both the "children functions" and the combined "predicate function" must be written to be compatible with JAX's vectorization scheme and just-in-time (JIT) compilation. This imposes a number of implementational constraints, most notably that the size and shape of all arrays must be fixed at compile time. This means, for instance, that the dimensions of the "legal action mask" (and, hence, the size of the action space in general) cannot change as the game progresses. In addition, values like the number of iterations in a loop or the positions of a lookup mask must essentially be "pre-specified." Crucially, however, values that are determined during *parsing* (such as the number

of children for a given node, or the value of any arguments) can be safely passed into compiled JAX functions as static constants. This fact is what allows `Ludax` to create JAX functions *dynamically* that nonetheless obey the constraints of vectorization and JIT compilation. At the top of the parse tree, these composed JAX functions are ultimately used to define the behaviors that appear in the environment's `step` function, such as applying the player's action to the board and handling move effects.

We next discuss some of the specific optimizations used by `Ludax`. In general, these are not *global* optimizations: they apply only to certain compositions of game rules and mechanics. Our approach is to deploy these optimizations when they are available and to "fall back" on slower but more general solutions when they are not.

**Precomputation:** An important optimization used by the `PGX` library (and JAX environments more generally) is to express functions as batched matrix operations rather than iterative procedures. For instance, rather than checking for a line of pieces in *Tic-Tac-Toe* by starting at the position of the last move and scanning out in each direction (as `Ludii`'s implementation does), `PGX` hard-codes the set of board indices that correspond to each possible line of three in the game (i.e. `[[0, 1, 2], [0, 3, 6], [0, 4, 7], ...]`) and performs a single multi-dimensional index into the board array – if any of the of the board index triples all correspond to positions occupied by a single player, then the game is over. `Ludax` adopts and generalizes this approach: during parsing of `line`, for example, the line indices are computed with respect to the size and shape of the game board (i.e. rectangular, hexagonal, ...) as well as the length and orientation of the desired line (i.e. diagonal, vertical, ...). Again, because these values depend only on attributes that are determined during parsing, they can be passed into JAX functions as constants. Precomputation naturally causes a trade-off between compile-time and run-time efficiency. In our case, we opt to use precomputation whenever possible, though some `masks` and `functions` cannot be expressed this way.

**Dynamic State Attributes:** Different games require tracking different kinds of information about the current game state. Most obviously, some games track a score for each player while others do not. When `Ludax` compiles a game, it automatically extracts the attributes required to instantiate a game state and omits the others, thereby reducing the memory footprint of the entire state object. More importantly, `Ludax` also automatically adds intermediary computations to each call of the environment's `step` function that help speed up later `mask`, `function`, or `predicate` evaluations. For example, in *Hex*, the game ends when one player manages to connect two opposite sides of the board with a continuous path of their pieces. Naively, checking whether the edges of the board are linked requires the expensive step of computing the board's connected components after each move. However, *updating* the board's connected components as a result of placing a single piece can be done very efficiently (a technique used well in the `PGX` implementation). At compile time, `Ludax` determines whether a game makes use of a "connection" rule and modifies the `step` function to iteratively update and track the board's connected components if so, greatly speeding up later checks. In future extensions, this functionality will be used to accommodate games with atypical or computationally expensive rules without affecting the runtime of existing games.

## 5 EXPRESSIVE RANGE

As mentioned above, `Ludax` currently supports a relatively narrow class of games: two-player, perfect-information board games played by placing, capturing, and sliding a single kind of game piece. Despite this, `Ludax`'s description language remains quite expressive. In addition to simple $m - n - k$ line completion games, `Ludax` supports complex win conditions (e.g. *misère* variants, score-based victory), asymmetric player goals, piece capturing and flipping, directional adjacency checks and restrictions, "custodial" mechanics, and games based on connecting arbitrary board regions. `Ludax` also supports regular rectangular and hexagonal boards of arbitrary sizes, as well as "hexagonal-rectangular" boards (e.g. as used in *Hex*). These components can then be combined compositionally to form a wide array of unique mechanics and dynamics. In addition, because `Ludax` is a general description language, implementing a single new game component expands the entire *space* of games in the framework. While the class of games representable in `Ludax` may at present be smaller than that of `Ludii` or other game description languages, it remains expansive. For example, `Ludax` is able to encode both *Yavalath* and *HopThrough* (see Figure 2) – games produced by the `Ludi` (Browne, 2009) and `GAVEL` (Todd et al., 2024) systems respectively, which were

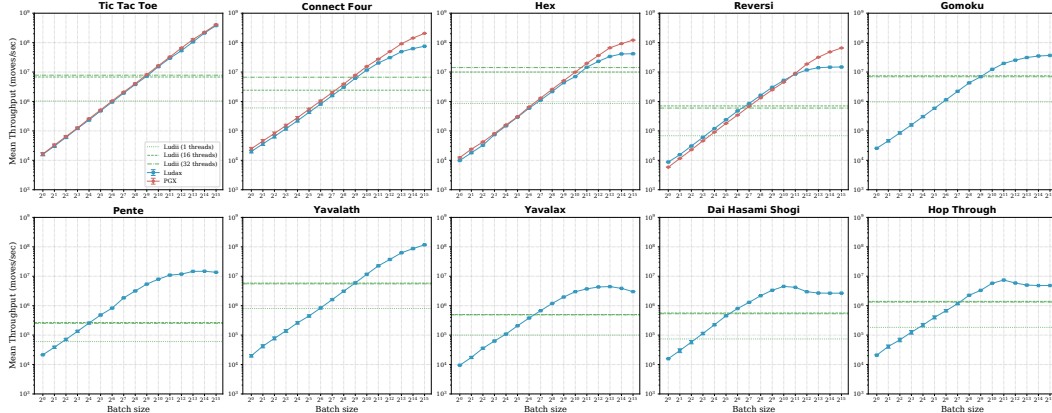

Figure 2: **Average throughput (moves per second) on various exemplar games for `Ludax`, `Ludii`, and `PGX`.** The first four games are implemented in all three frameworks, while the remaining games are implemented only in `Ludax` and `Ludii`. Speeds for `Ludax` and `PGX` are reported for 500 episodes of various batch sizes on a workstation with a single NVIDIA 4090 GPU and 32 CPU cores, while speeds for `Ludii` are reported for parallel execution on the same workstation across 1, 16, and 32 threads. Error bars are standard deviations calculated over the 500 episodes.

designed to automatically search through the space of games for novel exemplars. In Appendix E we detail a preliminary experiment on automatic game generation in `Ludax` via language models, where we find that two state-of-the-art open-weight LLMs were able to generate novel and potentially interesting games in `Ludax` without any finetuning or evolutionary search. This provides exciting initial evidence that `Ludax` is well-suited for game generation, and in future work it could serve as a meta environment for training general game-playing RL agents across the entire domain of expressible games.

# 6 BENCHMARKING

We benchmark the speed of `Ludax` on a set of 10 games, 4 of which are also implemented in both `Ludii` and `PGX` (allowing for a full comparison) and 6 of which are implemented only in `Ludax` and `Ludii`. Again, we emphasize that these 10 games are just *exemplars* of the class of games which `Ludax` supports, not an exhaustive list. A full description of each benchmark game is available in the Supplementary Material. We perform each of our benchmarking experiments on a workstation with a single NVIDIA 4090 GPU, 32 CPU cores, and 128GB of memory. In Figure 2 we plot the throughput (in steps per second) under a uniformly random action policy for each game environment against the batch size (log scale on both axes), with the standard deviation of throughputs across episodes as error bars. `Ludii` supports parallelization via multi-threading: we report throughput on the same workstation when parallelized on 1, 16, and 32 threads. Evaluations for `Ludax` and `PGX` were obtained by performing 100 warmup full-game episodes at the specified batch size, followed by measuring the speed over 500 episodes, with each evaluation taking at most a few minutes to complete. Evaluations for `Ludii` were obtained by running warmup episodes for 10 seconds, followed by measuring the speed over 30 seconds of episodes.[2] For games with potentially unbounded length (e.g. *Dai Hasami Shogi*), we terminate games for both `Ludax` and `Ludii` after 200 total turns.

Overall, `Ludax` achieves speeds that are competitive with state-of-the-art JAX environments. At small batch sizes, its throughput is similar to that of the `PGX` implementations. At larger batch sizes in more complicated games (i.e. *Hex* and *Reversi*), `PGX` takes a clear edge – though `Ludax` remains within an order of magnitude of `PGX`. The comparative "plateauing" of `Ludax`'s speed at high batch sizes may be due to memory pressure – for instance, `Ludax`'s implementation of *Hex* maintains both a board and the connected components for each game state, whereas the `PGX` implementation

---

[2]We opted to measure speed for `Ludax` and `PGX` using a fixed number of episodes because JAX's compilation procedure makes it difficult to halt execution after a specific elapsed wall time.

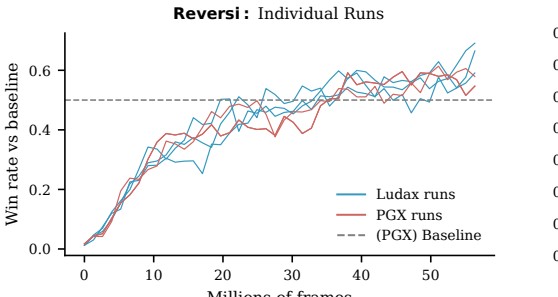 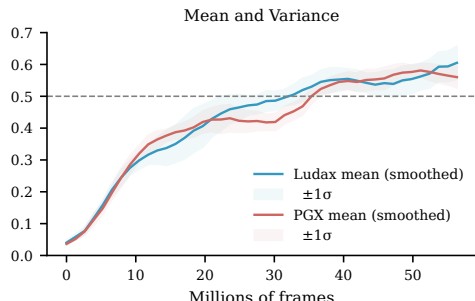

Figure 3: **Performance of reinforcement learning agents trained in the `Ludax` and `PGX` implementations of *Reversi* against the `PGX` baseline agent.** On the left, we plot the average winrate of the learned agents against the baseline over time and across three separate runs. On the right, we plot the average and variance of the winrates. Each run took roughly 3 hours to complete on a workstation with a single A100 GPU.

cleverly combines both into a single array. This kind of optimization is of course theoretically implementable in `Ludax` as well, though again we emphasize the desiderata of avoiding *game-specific* solutions.

`Ludax` also outspeeds `Ludii` on 16 and 32 threads across all 10 games, achieving a maximum speedup of between ~3x (*Hex*) and ~55x (*Pente*). We note that there are factors that both advantage and disadvantage `Ludax` in this specific comparison against `Ludii`. One potential advantage for `Ludax` is its smaller representation space – implementations of basic mechanics in `Ludii` support a wider range of optional arguments and board types, with a corresponding increase in computational overhead (though see Section 4 for how this may be avoided). Conversely, `Ludii`'s ability to use dynamically-sized data structures brings advantages that are particularly beneficial in uniformly random playouts, but would (partially) disappear in playouts using deep reinforcement learning. `Ludii` also has optimized playout implementations tailored towards many of the categories of games covered by `Ludax` (Soemers et al., 2022), though these optimizations are also more difficult to apply in the context of deep learning.

## 7 TRAINED AGENTS

Finally, we demonstrate the feasibility of training reinforcement learning agents using the `Ludax` framework. We train our agent on the game *Reversi* (also known as *Othello*) using the `AlphaZero`-style (Silver et al., 2017) training script from the `PGX` library[3] (making only slight modifications to accommodate minor differences between the `Ludax` and `PGX` APIs). We use the same ResNetV2 (He et al., 2016) network architecture and training hyperparameters as `PGX` (full details available in the Supplementary Material) and train three separate runs on a single A100 GPU. Each run lasted roughly 57 million frames and took roughly three hours to complete.

We compare the performance of agents trained in the `Ludax` and `PGX` environments against the baseline *Reversi* agent provided by the `PGX` library in Figure 3. Evaluations were performed by playing two batches of 1024 games (one with the learned agent as the first player and one as the second player), with actions sampled from the normalized output of the policy head at each step. We see that both learned agents achieve remarkably similar performances against the baseline, with little to no differences in learning speed or stability. While a more thorough, tournament-based evaluation would be necessary to properly rank the agents against each other, our objective is to demonstrate the general success of the training procedure and not to definitively defeat the baseline agent. Although the `PGX` implementation of the *Reversi* environment is slightly more efficient, this translated into only marginal improvements in overall runtime (about 1.5%) owing to the shared overhead of network forward passes and weight updates. Like `PGX`, `Ludax` offers a familiar API and an efficient set of implementations with which to train learned player agents.

---

[3]https://github.com/sotetsuk/pgx/blob/main/examples/alphazero/train.py (used under Apache 2.0 license)

## 8 LIMITATIONS

**Generality:** As mentioned in Section 5, `Ludax` currently supports a smaller class of games than other comparable game description languages. While we aim to increase the range of games expressible in `Ludax` (see below), it will likely never match the full generality of `Ludii`. As such, other frameworks may be more appropriate for use-cases in which a broad range of games is more important than rapid simulation. Further, `Ludax` does not support genres other than board games (e.g. video games, card games, ...) – we leave the development of hardware accelerated description languages for such domains as an exciting area of future work.

**Efficiency:** Compared to bespoke JAX implementations of board games (such as in the `PGX` library), environments in `Ludax` have slightly worse throughput – though the gap is marginal in a standard RL training setup. We deploy a number of optimizations to help close the efficiency gap when possible (see Section 4), but there are ultimately unavoidable trade-offs between speed and generality. For the purpose of training or benchmarking single-task agents on existing games, hard-coded simulators may remain the superior choice.

## 9 FUTURE WORK

The most obvious avenue of extension for `Ludax` is the implementation of additional game mechanics. In particular, we aim to support irregular board shapes, games with multiple piece types (e.g. *Checkers*) and games with multiple distinct gameplay phases (e.g. *Nine-Men's Morris*). In addition, it's also very likely that the implementation of specific gameplay elements could be further optimized for throughput and / or memory footprint. However, a balance must be struck between efficiency and generality: a less efficient solution which accommodates all valid games under the grammar is ultimately preferable to one which only applies to a subset of games. Lastly, we aim to provide a more robust visual interface for `Ludax`, both for the purpose of facilitating human-subject research (e.g. with packages like NiceWebRL(Carvalho et al., 2025)) and the potential development of more "human-like" artificial agents which process the game board at the pixel level and select actions spatially.

We are particularly excited about the potential application of `Ludax` to the study of *automated game design* (or reward-guided program synthesis more generally (Cui et al., 2021; Surina et al., 2025; Romera-Paredes et al., 2024)). Such systems depend on both a broad representation space and rapid evaluation of novel games – see Appendix E for a preliminary investigation of `Ludax`'s suitability for such research. The efficiency of `Ludax` may also make it possible to train a reinforcement learning agent from scratch as part of the inner loop of game evaluation, potentially unlocking a new range of computational features (e.g. learning curves) that correlate with human notions of fun and engagement. Relatedly, `Ludax` may prove useful to research on *human behavior and play*. Recent work has explored heuristic-based computational models of human play on simple line completion games (Zhang et al., 2024), and `Ludax` offers the possibility to both accelerate computation and broaden the domain to a wider class of games. Finally, `Ludax` offers an avenue to extend recent research in *general game playing* (e.g. with large language models (Schultz et al., 2024)) by providing a wide base of efficient game implementations that can in turn be leveraged for tree search algorithms or training world models.

## 10 CONCLUSIONS

We introduce a novel framework for games research that combines the generality of game description languages with the efficiency of modern hardware-accelerated learning environments. Our framework, `Ludax`, represents a broad class of two-player board games and compiles directly into code in the JAX Python library. Games in `Ludax` achieve speeds that are competitive with hand-crafted JAX implementations and faster than the widely-used `Ludii` game description language, and `Ludax` environments can easily be deployed in existing pipelines for deep reinforcement learning. Our framework helps widen and accelerate games research, with the potential to unlock new approaches in RL generalization, automatic game generation, and cognitive modeling.

ETHICS STATEMENT

This paper presents a general framework with the goal of advancing reinforcement learning and games research. While there are many potential societal consequences of such work in general, we do not feel that any must be specifically highlighted here. We emphasize that `Ludax` does not use or reproduce any copyrightable game material (i.e. art, specific expressions of rules, or game code). Low level game mechanics (such as those implemented in `Ludax`) are not copyrightable.

REPRODUCIBILITY STATEMENT

We provide the full source code for `Ludax` in the Supplemental Material in addition to the general dataset procedure in section 4. We provide the hyperparameters necessary to replicate our experiments in Appendix D and Appendix E

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

## A    EXAMPLE GAMES AND SYNTAX

Below we present the `Ludax` syntax for a small set of exemplar games (*Reversi*, *Connect Four*, *Yavalax*, and *Pente*) to help illustrate aspects of `Ludax`'s syntax and structure.

```
(game "Reversi"
    (players 2)
    (equipment
        (board (square 8))
    )

    (rules
        (start
            (place P1 (28 35))
            (place P2 (27 36))
        )
        (play
            (repeat (P1 P2)
                (place
                    (destination empty)
                    (result
                        (exists
                            (custodial any)
                        )
                    )
                    (effects
                        (flip (custodial any))
                        (set_score mover (count (occupied mover)))
                        (set_score opponent
                            (count (occupied opponent))
                        )
                    )
                )
                (force_pass)
            )
        )

        (end
            (if (passed both) (by_score))
)))
```

```
(game "Connect-Four"
    (players 2)
    (equipment
        (board (rectangle 6 7))
    )

    (rules
        (play
            (repeat (P1 P2)
                (place (destination (and
                        empty
                        (or
                            (edge bottom)
                            (adjacent occupied direction:up)
                        )
                    )))
                )
            )
        )

        (end
            (if (line 4) (mover win))
            (if (full_board) (draw))
        )
    )
)
```

```
(game "Yavalax"
    (players 2)
    (equipment
        (board (square 13))
    )

    (rules
        (play
            (once_through (P1)
                (place (destination empty))
            )
            (repeat (P2 P2 P1 P1)
                (place
                    (destination empty)
                    (result
                        (and
                            (not (line 5))
                            (not (= (line 4) 1))
                        )
                    )
                )
            )
        )
        (end
            (if (>= (line 4) 2) (mover win))
            (if (full_board) (draw))
        )
    )
)
```

```
(game "Pente"
    (players 2)
    (equipment
        (board (square 19))
    )

    (rules
        (play
            (once_through (P1)
                (place (destination center))
            )
            (repeat (P2 P1)
                (place
                    (destination empty)
                    (effects
                        (capture (custodial 2) increment_score:true)
                    )
                )
            )
        )
        (end
            (if (line 5) (mover win))
            (if (>= (score mover) 10) (mover win))
            (if (full_board) (draw))
        )
    )
)
```

Figure 4: **Ludax syntax for *Reversi* and *Connect Four* (classic board games), as well as *Yavalax* and *Pente* (modern board games).**

## B    LUDAX GRAMMAR

Below we present the complete grammar specification for `Ludax`, using the syntax of the `Lark` Python library (raw string constants omitted for brevity).

```
// ---Root---
game: "(game" name players equipment rules rendering? ")"
```

```
756   // ---Players---
757   players: "(players" positive_int ")"
758
759   // ---Equipment---
760   equipment: "(equipment" board")"
761   board: "(board" (board_square | board_rectangle | board_hexagon |
762      board_hex_rectangle) ")"
763   board_square: "(square" number ")"
764   board_rectangle: "(rectangle" number number ")"
765   board_hexagon: "(hexagon" number ")"
766   board_hex_rectangle: "(hex_rectangle" number number ")"
767
768   // ---Rules---
769   rules: "(rules" start_rules? play_rules end_rules ")"
770
771   // ---Start rules---
772   start_rules: "(start" start_rule+ ")"
773   start_rule: start_place
774   start_place: "(place" player_reference (pattern_arg |
775      multi_mask_arg) ")"
776
777   // ---Play rules---
778   play_rules: "(play" play_phase+ ")"
779   play_phase: phase_once_through | phase_repeat
780   phase_once_through: "(once-through" play_mover_order
781      play_super_mechanic ")"
782   phase_repeat: "(repeat" play_mover_order play_super_mechanic ")"
783   play_mover_order: "(" player_reference+ ")"
784
785   play_super_mechanic: play_mechanic force_pass?
786   play_mechanic: play_place | play_move
787   force_pass: "(force_pass" ")"
788
789   // ---Place rules---
790   play_place: "(place" mover_reference? place_destination_constraint
791      place_result_constraint? play_effects? ")"
792   place_destination_constraint: "(destination" super_mask ")"
793   place_result_constraint: "(result" super_predicate ")"
794
795   // ---Move rules---
796   play_move: "(move" move_types move_source_constraint
797      move_destination_constraint move_result_constraint?
798      play_effects? ")"
799   move_types: move_type | "(" move_type+ ")"
800   move_type: move_hop
801         | move_slide
802
803   move_hop: "hop" | "(hop" direction_arg ")"
804   move_slide: "slide" | "(slide" direction_arg ")"
805
806   move_source_constraint: "(source" super_mask ")"
807   move_destination_constraint: "(destination" super_mask ")"
808   move_result_constraint: "(result" super_predicate ")"
809
      // ---Effects---
      play_effects: "(effects" play_effect+ ")"
      play_effect: effect_capture
            | effect_flip
            | effect_increment_score
```

```
           | effect_set_score

effect_capture: "(capture" super_mask mover_reference? increment_
    score_arg? ")"
effect_flip: "(flip" super_mask mover_reference? ")"
effect_increment_score: "(increment_score" mover_reference
    function ")"
effect_set_score: "(set_score" mover_reference function ")"

// ---Functions---
function: function_add
        | function_connected
        | function_constant
        | function_count
        | function_line
        | function_multiply
        | function_score
        | function_subtract

function_add: "(add" function+ ")"
function_connected: "(connected" multi_mask_arg mover_reference?
    direction_arg? ")"
function_constant: positive_int
function_count: "(count" super_mask ")"
function_line: "(line" positive_int orientation_arg? exact_arg?
    exclude_arg? ")"
function_multiply: "(multiply" function+ ")"
function_score: "(score" mover_reference ")"
function_subtract: "(subtract" function function ")"

// ---End rules---
end_rules: "(end" end_rule+ ")"
end_rule: "(if" super_predicate end_rule_result ")"
?end_rule_result: result_win | result_lose | result_draw |
    result_by_score

// -- Result definitions --
result_win: "(" mover_reference "win" ")"
result_lose: "(" mover_reference "lose" ")"
result_draw: "(" "draw" ")"
result_by_score: "(" "by_score" ")"

// -- Mask definitions --
super_mask: mask | super_mask_and | super_mask_or | super_mask_not
super_mask_and: "(and" super_mask+ ")"
super_mask_or: "(or" super_mask+ ")"
super_mask_not: "(not" super_mask ")"

mask: mask_adjacent
    | mask_center
    | mask_column
    | mask_corners
    | mask_corner_custodial
    | mask_custodial
    | mask_edge
    | mask_empty
    | mask_occupied
    | mask_pattern
    | mask_prev_move
```

```
864        | mask_row
865

866   mask_adjacent: "(adjacent" super_mask direction_arg? ")"
867   mask_center: "center"
868   mask_column: "(column" positive_int ")"
869   mask_corners: "corners"
870   mask_corner_custodial: "corner_custodial" | "(corner_custodial"
871       mover_reference ")"
872   mask_custodial: "(custodial" custodial_length_arg mover_reference?
873        orientation_arg? ")"
874   mask_edge: "(edge" edge ")"
875   mask_empty: "empty"
876   mask_occupied: "occupied" | "(occupied" mover_reference ")"
877   mask_pattern: "(pattern" dimensions_arg pattern_arg rotate_arg? ")
          "
878   mask_prev_move: "(prev_move" mover_reference ")"
879   mask_row: "(row" positive_int ")"
880

881   multi_mask: multi_mask_corners
882            | multi_mask_edges
883            | multi_mask_edges_no_corners
884

885   multi_mask_corners: "corners"
886   multi_mask_edges: "edges"
887   multi_mask_edges_no_corners: "edgesNoCorners"
888

889   // ---Predicate definitions---
890   super_predicate: predicate | super_predicate_and |
891       super_predicate_or | super_predicate_not
892   super_predicate_and: "(and" super_predicate+ ")"
893   super_predicate_or: "(or" super_predicate+ ")"
894   super_predicate_not: "(not" super_predicate ")"
895
896   predicate: predicate_equals
897           | predicate_exists
898           | predicate_full_board
899           | predicate_function
900           | predicate_greater_equals
901           | predicate_less_equals
902           | predicate_mover_is
903           | predicate_passed
904
905   predicate_equals: "(=" function+ ")"
906   predicate_exists: "(exists" super_mask ")" // technically
907       equivalent to (>= (count mask) 1)
908   predicate_full_board: "(" "full_board" ")"
909   predicate_function: function // special syntax which is equivalent
910        to "(>= function 1)"
911   predicate_greater_equals: "(>=" function function ")"
912   predicate_less_equals: "(<=" function function ")"
913   predicate_mover_is: "(mover_is" player_reference ")"
914   predicate_passed: "(passed" (mover_reference | BOTH) ")"
915
916   // Additional (potentially optional) arguments for predicates
917   custodial_length_arg: ANY | positive_int
      dimensions_arg: "(" positive_int positive_int ")"
      direction_arg: "direction:" direction
      exact_arg: "exact:" boolean
      exclude_arg: "exclude:" multi_mask_arg
```

```
918  increment_score_arg: "increment_score:" boolean
919  multi_mask_arg: multi_mask | super_mask | "(" super_mask+ ")"
920  orientation_arg: "orientation:" orientation
921  pattern_arg: "(" positive_int+ ")"
922  rotate_arg: "rotate:" boolean
923
924  // Optional rendering details
925  rendering: "(rendering" rendering_detail+ ")"
926  rendering_detail: color_assignment
927
928  color_assignment: "(color" player_reference color ")"
929
930  // General-purpose definitions
930  ?number: SIGNED_NUMBER
931  ?positive_int: /[0-9]+/
932  ?boolean: TRUE | FALSE
933  ?edge: TOP | BOTTOM | LEFT | RIGHT | TOP_LEFT | TOP_RIGHT |
934      BOTTOM_LEFT | BOTTOM_RIGHT
935  ?direction: UP | DOWN | LEFT | RIGHT | UP_LEFT | UP_RIGHT |
936      DOWN_LEFT | DOWN_RIGHT | VERTICAL | HORIZONTAL | ORTHOGONAL |
937      DIAGONAL | BACK_DIAGONAL | FORWARD_DIAGONAL | ANY
938  ?orientation: VERTICAL | HORIZONTAL | ORTHOGONAL | DIAGONAL |
939      BACK_DIAGONAL | FORWARD_DIAGONAL | ANY
940  ?color: WHITE | BLACK
940  // --------------------------
941
942  ?player_reference: P1| P2
943  ?mover_reference: MOVER | OPPONENT
944  name: STRING
945  variable_name: /\?[a-z][a-z0-9]*/
946  id: /[a-zA-Z0-9_]+/
```

## C  BENCHMARK GAME DESCRIPTIONS

Below, we present natural language descriptions of the rules for each of the exemplar games analyzed in Section 6.

***Tic-Tac-Toe***: Players take turns placing a piece into an empty space on a square 3-by-3 board. If a player forms a line of three of their pieces in a row (either vertically, horizontally, or diagonally), they win. If the board is completely full but no lines have been formed, then the game ends in a draw.

***Connect Four***: Players take turns placing a piece into the top of one of the seven columns on a 6-by-7 board. The piece then "falls" until it rests on either the bottom of the board or another piece. A player can't place a piece into a column that is already "full." If a player forms a line of four of their pieces in a row (either vertically, horizontally, or diagonally), they win. If the board is completely full but no lines have been formed, then the game ends in a draw.

***Hex***: Players take turns placing a piece into an empty space on an 11-by-11 board composed of hexagonal tiles (forming a parallelogram, see visual depiction here). The objective for the first player is to form a continuous path of their pieces that connects the top edge of the board with the bottom edge, while the objective for the second player is to do the same but connect the left and right edges of the board. The first player to achieve their objective wins the game. Because of the geometric properties of the board, it's not possible for the game to end in a draw.

***Reversi***: The game takes place on a square 8-by-8 board. To begin, a white piece is placed at positions D4 and E5 and a black piece is placed at positions D5 and E4 (see visual depiction here). Players take turns placing a piece into an empty space such that a line of one or more of the opponent's pieces are "sandwiched" on either end by the player's pieces. This configuration is called a "custodial" arrangement of pieces. After placing a piece, any of the opponent's pieces which are in

such a custodial arrangement are flipped and now belong to the player who just moved. It's possible for a single move to form multiple custodial arrangements in different directions, in which case all of the relevant pieces are flipped. If a player cannot make a legal move, they must pass (and they cannot pass without making a move otherwise). If both players pass, then the game is over. The winner is determined by the player who has the largest number of pieces on the board at the end of the game (in the event of a tie, the game ends in a draw).

***Gomoku***: Players take turns placing a piece into an empty space on a square 15-by-15 board. If a player forms a line of exactly five of their pieces in a row (either vertically, horizontally, or diagonally), they win. However, forming a line of six or more does not count – the player must have at least one line of exactly five. If the board is completely full but no lines of exactly five have been formed, then the game ends in a draw.

***Pente***: Players take turns placing a piece into an empty space on a square 19-by-19 board. If a player forms a line of five of their pieces in a row (either vertically, horizontally, or diagonally), they win. In addition, if placing a piece causes a line of exactly two of the opponent's pieces to be put into a custodial arrangement, the two pieces are captured and removed from a board. Note that placing a piece *into* a custodial arrangement formed by the opponent does not result in any pieces being captured. A player who captures at least 10 of the opponent's pieces over the course of the game wins. In the variant of *Pente* implemented in `Ludii` and `Ludax`, the first player must make their first move into the exact center of the board.

***Yavalath***: Players take turns placing a piece into an empty space on a regular hexagonal board with a diameter of 9 spaces. If a player forms a line of four of their pieces in any direction (either diagonally or horizontally[4]), they win. However, if a player forms a line of three of their pieces in a row without also forming a line of four, they lose. If the board is completely full but no lines of four or three have been formed, then the game ends in a draw.

***Yavalax***: To begin, the first player places a piece into an empty space on a square 13-by-13 board. Starting with Player 2, players then take turns placing two pieces into empty spaces on the board. If a player forms at least two distinct lines of four of their pieces in any direction (either vertically, horizontally, or diagonally), they win. However, a player may not place a piece into a space if doing so would form a line of five pieces in any direction or if it would form exactly one line of four pieces in any direction. Note that this restriction applies to a player's first move of their turn even if they could form a second line of four pieces with their second move of the turn (and thus win). If the board is completely full and neither player has formed at least two distinct lines of four pieces, then the game ends in a draw.

***Dai Hasami Shogi***: The game takes place on a square 9-by-9 board. To begin, white pieces are placed on the bottom two rows of the board and black pieces are placed on the top two rows. Players take turns moving one of their pieces, either by sliding it any number of squares vertically or horizontally (i.e. as a rook) or by hopping over one piece (belonging to either player) vertically or horizontally into an empty square. Hopping over a piece does not capture it, but opposing pieces can be captured "custodially" (i.e. by moving to surround an enemy piece on both sides vertically or horizontally). An opponent's piece in a corner can also be captured by moving a piece to occupy both orthogonally-adjacent squares. A player wins if they manage to form a horizontal or vertical line of 5 pieces in a row if none of those pieces are in their starting rows.

***HopThrough***: The game takes place on a square 8-by-8 board. To begin, white pieces are placed on the bottom two rows and black pieces are placed on the top two rows. Players take turns moving one of their pieces by hopping over an adjacent piece (belong to either player) in any direction. Hopping over a piece does not capture it. A player wins if they manage to get one of their pieces to the opposite edge of the board (i.e. the top edge for the first player and the bottom edge for the second player).

---

[4]`Ludax` assumes a canonical orientation for hexagonal boards in which the diameter stretches from left to right, though it is functionally equivalent to the orientation in which the diameter runs vertically)

```
Invent simple rules for a novel two player abstract strategy game
called {name}. Implement it in the ludax language. You will find
attached the ludax's grammar as well as a few examples of games
implemented in ludax. Start by implementing a simplified version
of your rules, and then incrementally add rules that are harder to
express in ludax. At each step, make sure you write a compilable
game according to ludax's grammar.
```

Listing 1: System instruction for LLM-based generation.

## D   TRAINING HYPERPARAMETERS

Below we provide the exact training hyperparameters used in the reinforcement learning experiments in Section 7. These are largely copied from the `PGX` implementation.

- **Model architecture:** `ResnetV2`
- **Number of channels:** 128
- **Number of layers:** 6
- **Self-play batch size:** 1024
- **Self-play simulations:** 32
- **Self-play max steps:** 256
- **Training batch size:** 4096
- **Learning rate:** 0.001
- **Evaluation frequency:** 5
- **Training iterations:** 219

Note that each "iteration" consists of generating play data for 256 steps using the self-play batch size of 1024 (see Koyamada et al. (2023)). We train the model for 219 iterations, which corresponds to $256 \times 1024 \times 219 = 57409536$ (or roughly 57 million) steps in the environment.

## E   GAME GENERATION

We attempt to synthesize new games in the `Ludax` DSL using two approaches: random sampling and LLM-based generation. In Table 1, we present the `GAVEL` game evaluation metrics for each method.

**Random Sampling:** Games are generated by naive uniform random sampling. Starting from the root game "ludeme" (i.e. production rule), we sample the next ludeme among those which are valid continuations according to the grammar. Additionally, we impose a maximum syntax tree depth of 5, beyond which a closing bracket is always given priority.

**LLM-based Generation:** Games are generated as a few-shot task. The model is prompted with a system instruction (Listing 1), the full grammar (Appendix B), and the game implementations from Appendix C as examples. The model is instructed to describe the rules of a new game and produce multiple `Ludax` implementations of increasing complexity; we evaluate only the final game produced. To encourage diversity, each attempt is seeded with a randomly generated and nonsensical game name such "Outstanding Rainbow Spaniel."

**GAVEL-like Evaluation:** Inspired by Todd et al. (2024), we assess each generated game as follows:

1. A game is *playable* if its description compiles and runs without error.

2. For each *playable* game, we run agent-vs-agent playthroughs using a custom JAX implementation of MCTS with UCB1 (Kocsis & Szepesvári, 2006)

3. We compute the following heuristics from these playthroughs:

- **Balance:** max winrate gap between players
- **Decisiveness:** fraction of non-draw outcomes
- **Completion:** fraction of games reaching a terminal state
- **Agency:** fraction of turns with > 1 legal move
- **Coverage:** fraction of board sites occupied at least once
- **Strategic Depth:** difference in winrate between a stronger MCTS agent and a weaker one (fewer simulations).

The overall "`GAVEL` score" is the harmonic mean of the individual heuristic scores. Games with a `GAVEL` score > 0.4 are deemed potentially *interesting*. We note that this experiment is preliminary: it omits diversity measures, and the limited search budget for MCTS means they will frequently miss good moves a stronger agent might find. Nevertheless, the fact that an LLM can implement novel games in `Ludax` without finetuning suggests that `Ludax` 's grammar is intuitive and highlights its potential for both game generation.

**Hyperparameters:** For each method, we sample 100 games. For the LLM-based methods, we use a sampling temperature of 0.2. To compute the evaluation score, we run 100 agent-vs-agent simulations for each game. The MCTS agents perform 100 iterations (i.e. traversal, expansion, and random rollout) for each action. For the "strategic depth" evaluation we compare against an MCTS agent that performs 50 iterations per action.

Table 1: GAVEL-based evaluation metrics for 100 generated games, obtained either by uniform random sampling or an LLM. As a baseline, we report results for all default games in Appendix C. *Playable* and *Interesting* denote percentages over all generated games (*Playable ≥ Interesting*). *GAVEL score* and *Strategic Depth* report the median and standard deviation, computed only on playable games.

| Method | Playable | Interesting | GAVEL Score | Strategic Depth |
|---|---|---|---|---|
| Default Games | 100% | 100% | 0.69 ±0.15 | 0.66 ±0.15 |
| Random Sampling | 4% | 0% | 0.00 ±0.00 | 0.00 ±0.00 |
| GPT-OSS-120B | 95% | 83% | 0.59 ±0.22 | 0.58 ±0.17 |
| LLaMa-4-17B | 82% | 42% | 0.49 ±0.21 | 0.68 ±0.23 |

