# OpenReview forum: "Ludax: A GPU-Accelerated Domain Specific Language for Board Games"
_ICLR.cc/2026/Conference — Submitted to ICLR 2026_

### Official Review · Reviewer_tBHe · 2025-10-26

**Soundness:** 2
**Presentation:** 2
**Contribution:** 2
**Rating:** 4
**Confidence:** 4

**Summary:**

The authors propose Ludax, a framework that automatically generates vectorizable JAX environments for board games.
It introduces a dedicated game description language (DSL) and a pipeline that converts this DSL into executable JAX code.
The authors also highlight several design techniques for maintaining JAX jittability—ensuring static shapes and pure functional style.
In experiments with selected board games, Ludax-generated environments achieved comparable speed to hand-crafted JAX baselines and significantly outperformed the non-vectorized Java-based framework Ludii, from which Ludax draws inspiration.

**Strengths:**

While neither DSL-based game generation nor vectorized environments are entirely new concepts, their combination is valuable and timely.
Conceptually, it matters because recent advances in reinforcement learning algorithmsrely heavily on vectorizable simulators such as PQN, and technically, because implementing diverse games in JAX is non-trivial due to strict requirements for jittability (static array shapes, pure functional programming).

Ludax effectively addresses these challenges by offering a DSL and a code generation pipeline tailored for JAX.
The paper clearly explains how to define atomic functions, map them to DSL primitives, and generate executable JAX code.
The experiments convincingly show that the generated code is both fast and compatible with RL training pipelines, as evidenced by the consistent learning performance compared to manually written environments.

**Weaknesses:**

As the authors themselves note, limited generalizability is the main bottleneck.
To define a new game, one must manually implement new atomic functions that are both compositional and general enough to be reused for other games.
However, many game rules are highly domain-specific and cannot be easily decomposed into such atomic components.

Thus, Ludax excels at interpolating within the family of games expressible by existing atomic functions (e.g., two-player, rectangular board, placing or sliding pieces, line-based victory conditions such as Hex, Gomoku, Connect Four, or Yavalath).
But it struggles to extrapolate to more complex games requiring novel logic.
Although the authors mention that adding new atomic functions can extend Ludax, designing such composable primitives remains a core research challenge rather than an engineering task.
Hence, this limitation should not be overlooked as a minor or easily solvable issue.

**Questions:**

- Do the authors have any insight into how to incorporate complex, non-compositional game logic into Ludax’s atomic framework? Designing such reusable primitives seems to require substantial domain intelligence.

- As a concrete example, could the authors show how Ludax could be extended to implement a more complex game like Go, which lies close to its current scope but introduces challenging mechanics such as group connectivity, liberties, and ko rules?

---

> ### Author Response · Authors · 2025-11-20
>
> We thank the reviewer for their encouraging feedback, noting that the work is both “valuable and timely.” We take the feedback of Ludax’s “main bottleneck” seriously, and hope that recent expansions to the representation space (as well as a discussion of the feasibility of further expansions) goes some way towards addressing these concerns.
>
> **Domain Expansion:** In the time since the initial submission, we have continued to expand the range of games which Ludax supports. Most notably, we have added support for games with multiple distinct piece types. This opens up an entire new space of possible games, including asymmetric games (such as Wolf and Sheep) and games with promotion mechanics (such as Checkers). As part of implementing American Checkers, we have also greatly expanded the range of low-level mechanics Ludax supports. Ludax now allows designers to specify “conditional effects” (e.g. “if a player makes a capture, then they can move again”) as well as “move priorities” that determine a hierarchy of legal actions (e.g. “pieces can either slide or capture by hopping, but if a capture is legal the player must take it”). We also added support for “relative directions” that allow game descriptions to more compactly specify the movement of pieces that go “forwards” and “backwards.”  This major refactor also sets the stage for further expansions, such as the introduction of “neutral” pieces (as in Amazons). While Ludax already supported a broader space of games than comparable JAX-based frameworks, we feel that this recent expansion helps solidify it as a general-purpose representation scheme for board games.
>
> **Ease of Further Expansions:** The challenges to supporting more games in Ludax are largely mundane. Implementing new low-level mechanics requires writing and optimizing JAX code and testing the results to make sure they integrate properly with the rest of the language. While this work is non-trivial, it is also not exceptionally difficult. As mentioned above, we have already made a significant expansion to the space of games expressible in Ludax and will continue to do so into the future. Further, there is a “scaffolding” effect where implementing new games becomes easier the more mechanics have already been implemented. For instance, adding support for Checkers required implementing a number of novel mechanics (promotion, forced captures, extra turn restrictions, ...), but now those mechanics can be adapted for other, related games without requiring any additional implementation efforts. We will include a more robust version of this discussion in our revision of the manuscript.
>
> **Feasibility of Go:** Go is an excellent example of a game which is outside of Ludax’s current representation space but does not necessarily impose any theoretical barriers in terms of implementation. At a low level, determining group connectivity is very related to determining the board’s connected components, which is already implemented as part of Hex (and related games). More generally, the PGX library has demonstrated that it’s fully possible to write an efficient implementation of Go in JAX. A theoretical Ludax implementation of Go would not need to re-invent any implementation details -- instead it would be a matter of generalizing the implementation (e.g. to other board sizes and shapes) and attaching the mechanics to the appropriate syntactic components of the language.

---

> > ### Comment · Reviewer_tBHe · 2025-11-24
> >
> > Thank you very much for your detailed reply.
> > Your explanation has alleviated some of my concerns regarding the feasibility of expanding the range of supported games. In particular, I acknowledge that we will benefit from the scaffolding effect as more games and mechanics are added to Ludax.
> >
> > However, at this stage I would still prefer to stick the current evaluation. The main reason is that, even with the scaffolding effect, there will likely always be game-specific logic that is difficult to implement and requires substantial human effort (in this moment, the atomic logic is no sufficient). Furthermore, Ludax’s current interpolation ability still appears to be limited: there is no instantiation beyond trivial expansion.
> >
> > That said, I consider this line of research to be important and promising, and I believe it merits publication in the future. I hope the authors will further develop this work to cover a wider variety of game logics and to demonstrate more striking interpolation capabilities.

---

> > > ### Author Response · Authors · 2025-11-25
> > >
> > > Thank you for the response! We're glad that we've addressed some of your concerns. For our own benefit, we're curious if you could expand a bit more on what you mean by "interpolation" and "no instantiation beyond trivial expansion." Ludax already allows you to create games that cross disparate mechanics (such as a Yavalath variant that also incorporates the connectivity component of Hex, or a variant of Wolf and Sheep with custodial capture mechanics). Is there another kind of mechanical crossover you would like to see?

---

### Official Review · Reviewer_ABz8 · 2025-11-03

**Soundness:** 3
**Presentation:** 2
**Contribution:** 2
**Rating:** 4
**Confidence:** 3

**Summary:**

In this paper, the authors present a DSL, Ludax, for board games that combines the expressiveness of Ludii with the GPU performance of PGX. Ludax's design draws heavily on the Ludii game description DSL, but changes are introduced to improve performance and simplify the description of certain meaningful game states. Ludax game descriptions are compiled using the Lark Python library, and operators are processed from the bottom up into atomic JAX functions. Some of these JAX functions are converted into maps from the current game state into a Boolean truth value and passed up the parse tree. A major improvement of Ludax is the transformation of iterative procedures related to game states into matrix-value functions amenable to fast precomputation using GPUs, similar to PGX. The authors support the efficiency of Ludax in the evaluation section by comparing the performance with PGX and achieving comparable results while maintaining similar convergence trajectories during training.

**Strengths:**

- Ludax combines the representation power of Ludii for board games with the performance improvements of a bespoke implementation written by hand in PGX. This strategy will allow researchers to investigate a wide range of existing board game dynamics quickly and implement new games amenable to accelerated training.
- The performance profiles comparing Ludii, Ludax, and PGX illustrate nicely the level of training throughput achievable using Ludax, as shown in Figure 2.
- The fidelity of the game dynamics after lowering to JAX using Ludax is illustrated in Figure 3 with very similar training dynamics when using Ludax and PGX.

**Weaknesses:**

- The biggest weakness is the lack of breadth of the current features of Ludax compared to Ludii. Although Ludax does provide support several games, the contribution would be significantly higher if more environments were supported, given the number of games available in Ludii.
- The presentation of Ludii in Section 3 is extensive, but an extensive portion is dedicated to describing existing components instead of focusing on the contributions of Ludax.
- This paper seems like an interesting direction, but the current limitations make it fall short of the contribution threshold for the current venue, in my opinion.

**Questions:**

- What are the primary hurdles impeding the support for more Ludii environments?

---

> ### Author Response · Authors · 2025-11-20
>
> We thank the reviewer for their analysis and feedback. We hope that recent expansions to Ludax’s representation space and the additional clarifications below help address some of their concerns.
>
> **Domain Expansion:** In the time since the initial submission, we have continued to expand the range of games which Ludax supports. Most notably, we have added support for games with multiple distinct piece types. This opens up an entire new space of possible games, including asymmetric games (such as Wolf and Sheep) and games with promotion mechanics (such as Checkers). As part of implementing American Checkers, we have also greatly expanded the range of low-level mechanics Ludax supports. Ludax now allows designers to specify “conditional effects” (e.g. “if a player makes a capture, then they can move again”) as well as “move priorities” that determine a hierarchy of legal actions (e.g. “pieces can either slide or capture by hopping, but if a capture is legal the player must take it”). We also added support for “relative directions” that allow game descriptions to more compactly specify the movement of pieces that go “forwards” and “backwards.”  This major refactor also sets the stage for further expansions, such as the introduction of “neutral” pieces (as in Amazons). While Ludax already supported a broader space of games than comparable JAX-based frameworks, we feel that this recent expansion helps solidify it as a general-purpose representation scheme for board games.
>
> **Novelty with Respect to Ludii:** As a minor clarification, section 3 does not describe Ludii -- it describes Ludax. Ludax takes inspiration from Ludii in terms of how (in very general terms) it describes game rules, but the syntax is entirely bespoke and designed to accommodate both the constraints of working in JAX and the requirements of modern deep learning pipelines. Section 3.4 highlights some philosophical differences between Ludii and Ludax, but it is far from the only way in which they are distinct. We will make this point clearer in our revision of the manuscript.
>
> **Ease of Further Expansions:** The challenges to supporting more games in Ludax are largely mundane. Implementing new low-level mechanics requires writing and optimizing JAX code and testing the results to make sure they integrate properly with the rest of the language. While this work is non-trivial, it is also not exceptionally difficult. As mentioned above, we have already made a significant expansion to the space of games expressible in Ludax and will continue to do so into the future. Further, there is a “scaffolding” effect where implementing new games becomes easier the more mechanics have already been implemented. For instance, adding support for Checkers required implementing a number of novel mechanics (promotion, forced captures, extra turn restrictions, ...), but now those mechanics can be adapted for other, related games without requiring any additional implementation efforts. We will include a more robust version of this discussion in our revision of the manuscript.

---

### Official Review · Reviewer_d2bZ · 2025-11-03

**Soundness:** 4
**Presentation:** 4
**Contribution:** 2
**Rating:** 2
**Confidence:** 3

**Summary:**

This paper provides for a jax accelerated RL framework for games. Ludax is a domain specific language for games, that allows for the description of  hundreds of games, and then is automatically optimized for Jax.

**Strengths:**

The paper is well written, and the motivations are clear. Games are certainly important in the history of AI, and have led to many breakthroughs. Further, a language like Ludax has uses beyond games, and can express a variety of problems, and also be useful in analyzing RL generation. or rapid testing on procedurally generated game environments. Impressively, the speed of Ludax games is comparable with specific game jax implementations. The supplemental materials were clear, and the code was easy to review.

**Weaknesses:**

My main issue with the paper is its limited novelty. While section 3.4 does discuss some non trivial differences from Ludii, fundamentally Ludax is simply a Jax port of Ludii. While this is certainly a very useful achievement, and I am sure it will be used, I do not believe that slightly modifying an existing tool to work with Jax merits a paper at ICLR. I commend the authors for their quality work. and recommend submitting to a more appropriate venue.

**Questions:**

I would ask the authors to elaborate on the novelty of this work, beyond reworking some parts of Ludii to work with Jax (section 3.4).

---

> ### Author Response · Authors · 2025-11-20
>
> We thank the reviewer for their consideration and encouraging feedback, and hope to clarify some points about Ludax’s novelty with respect to prior work.
>
> Simply put, Ludax is not just a re-tooling of an existing description language. It was designed and implemented from the ground up to work within the tight constraints of programming in JAX (i.e. no conditional logic, static shapes) and to also slot into existing deep learning pipelines (something previous game description languages do not support). The Ludii language was used as a high-level inspiration for the design of the description language, but we did not use or rework any of its implementations.
>
> In addition, we view one of the primary contributions of Ludax as being the introduction of a new paradigm for combining JAX with RL environment description languages. To our knowledge, we are the first to describe the compositional syntax tree procedure in which low-level transformation functions (i.e. mappings from states to masks, booleans, and scalar values) are combined into higher-order transformations and eventually into interactive environments. While Ludax uses this technique to compile board game rules, the general approach is broad enough to support a variety of other domains. We feel that the combination of a variety of new game implementations in JAX (i.e. of existing games not in PGX and of game variants) and the introduction of a new paradigm for integrating JAX into description languages makes Ludax a valuable contribution to the datasets and benchmarks track.

---

### Official Review · Reviewer_UUTF · 2025-11-06

**Soundness:** 3
**Presentation:** 2
**Contribution:** 3
**Rating:** 4
**Confidence:** 3

**Summary:**

The paper presents Ludax, a domain-specific language (DSL) for describing two-player, perfect-information board games that compiles into GPU-accelerated JAX environments. It combines the human-readable syntax inspired from Ludii with PGX, enabling rapid simulation and model training across a class of placement/capture/connection games for LLMs. The compilation process transforms high-level game rules into composable, JIT-compilable JAX functions with optimizations such as precomputed line indices and dynamic state tracking. The conducted experiments show significant throughput improvement compared to Ludii and demonstrate the capability of RL agent training.

**Strengths:**

1. Novelty: Ludax successfully combine PGX with Ludii to allow them compliment to each other and achieve both generality and acceleration via principled code generation.

2. Usability: The environment is able to directly fit into JAX-based RL pipelines and the game descriptions closely mirror natural language lowering the technical requirements for domain experts and researchers to prototype variants. The scalability and structured description make it possible to train LLM for game generation, reasoning and potentially world modeling.

**Weaknesses:**

1. Hard Limitation: Limited to single-piece, placement/capture games on regular boards. No support for multi-piece types, stacking, promotion, or irregular geometry.

2. Benchmark Analysis: No memory profiling or compile-time analysis. No ablation of optimizations (precompute vs. naive). No large-board stress test (e.g., 19×19 Pente).

2. LLM integration: No demonstration of LLM-guided search over Ludax space (e.g., evolving win conditions) or RL generalization across generated variants.

**Questions:**

Address weakness above.

---

> ### Author Response · Authors · 2025-11-20
>
> We thank the reviewer for their consideration and feedback. We hope that some of the progress in Ludax and the additional clarifications below helps address their concerns.
>
> **Domain Expansion:** In the time since the initial submission, we have continued to expand the range of games which Ludax supports. Most notably, we have added support for games with multiple distinct piece types. This opens up an entire new space of possible games, including asymmetric games (such as Wolf and Sheep) and games with promotion mechanics (such as Checkers). As part of implementing American Checkers, we have also greatly expanded the range of low-level mechanics Ludax supports. Ludax now allows designers to specify “conditional effects” (e.g. “if a player makes a capture, then they can move again”) as well as “move priorities” that determine a hierarchy of legal actions (e.g. “pieces can either slide or capture by hopping, but if a capture is legal the player must take it”). We also added support for “relative directions” that allow game descriptions to more compactly specify the movement of pieces that go “forwards” and “backwards.”  This major refactor also sets the stage for further expansions, such as the introduction of “neutral” pieces (as in Amazons). While Ludax already supported a broader space of games than comparable JAX-based frameworks, we feel that this recent expansion helps solidify it as a general-purpose representation scheme for board games.
>
> **Further Benchmarking:** Game environments in Ludax take on the order of a few seconds to compile. While this is longer than static environments that don’t incorporate an underlying grammar, it is negligible for the primary use case of a JAX-based environment (namely, running millions of steps on a single task). We’re happy to collect exact statistics on compilation time if the reviewer wishes.
>
> We’d also like to point out that the version of Pente reported in the paper is the 19x19 board variant (see Appendix C for a full description). Our implementation runs at over 10 million steps per second and is more than 50 times faster than Ludii on multiple threads, indicating that Ludax can indeed scale to large board sizes.
>
> **LLM Experiments:** We present an initial experiment on LLM-guided generation in Appendix E, where we show that LLMs are indeed capable of producing potentially-interesting board games using Ludax’s syntax. We agree that LLM-based game generation is an exciting avenue for future work, and our initial results indicate that Ludax is an appropriate choice of representation format for such research. We will signpost these findings more clearly in the main text of our manuscript.

---

### Author Response · Authors · 2025-12-02

We'd like to once again thank the reviewers for their consideration of our work and their helpful comments. Multiple reviewers indicated some hesitance about Ludax's expressive range: we hope that the expansion of the underlying grammar to include potentially asymmetric games with multiple piece types and complex move hierarchies (e.g. Checkers) helps address these concerns. We also hope that we have successfully clarified some points regarding our experiments (i.e. that the paper includes a large-board benchmark and initial LLM experiments) and novelty (i.e. that Ludax does not simply reimplement an existing game description language in JAX). Again, we appreciate the feedback and the opportunity to strengthen our work.

---

### Meta-Review · Area_Chair_P5dH · 2026-01-06

**Summary:**

This paper presents Ludax, a game description language that draws heavy inspiration from Ludii but is implemented in JAX. This allows researchers to define games that can be simulated with GPU-acceleration, enabling faster iteration in RL research.

The primary concern raised by reviewers is the generality of Ludax compared to Ludii. Mapping game logic to efficient JAX is a complicated task and this limits the number of games that can be expressed in Ludax. While the authors have expanded Ludax since the initial submission, I do not believe this expansion is yet sufficient to alleviate this concern.

I sincerely encourage the authors to continue working on this. This seems like a promising direction and the concern about generality is one that will go away as Ludax is continued to be expanded (even if it never achieves the full breadth of Ludii).

**Reviewer Concerns:**

## Reviewer UUTF

- The types of games that can be expressed are limited. I believe this concern is still outstanding.
- Lack of benchmark analysis and extensiveness. I believe this concern is addressed.
- No demonstration of llm integration. I believe this is addressed.

## Reviewer d2bZ

- The novelty of Ludax. I largely disagree with this concern (and it is not echoed in any other reviews) and thus believe this concern was sufficient addressed.

## Reviewer ABz8

- Lack of breadth of games that can be realized in Ludax. I believe this concern is still outstanding.

## Reviewer tBHe

- The breadth of games that can be realized in Ludax. I believe concern is still outstanding.

**Reviewer Scores:**

I do not believe the reviewer scores would have meaningfully changed after rebuttal.

---

### Decision · Program_Chairs · 2026-01-26

Reject